# Carboxamide Derivatives Are Potential Therapeutic AHR Ligands for Restoring IL-4 Mediated Repression of Epidermal Differentiation Proteins

**DOI:** 10.3390/ijms23031773

**Published:** 2022-02-04

**Authors:** Gijs Rikken, Noa J. M. van den Brink, Ivonne M. J. J. van Vlijmen-Willems, Piet E. J. van Erp, Lars Pettersson, Jos P. H. Smits, Ellen H. van den Bogaard

**Affiliations:** 1Department of Dermatology, Radboud Institute for Molecular Life Sciences (RIMLS), Radboud University Nijmegen Medical Center (Radboudumc), 6525 GA Nijmegen, The Netherlands; gijs.rikken@radboudumc.nl (G.R.); noa.vandenbrink@radboudumc.nl (N.J.M.v.d.B.); ivonne.vanvlijmen-willems@radboudumc.nl (I.M.J.J.v.V.-W.); piet.vanerp@radboudumc.nl (P.E.J.v.E.); jos.ph.smits@radboudumc.nl (J.P.H.S.); 2Immunahr AB, Prästasvängen 21, 224 80 Lund, Sweden; larspettersson59@live.se

**Keywords:** aryl hydrocarbon receptor, carboxamide-3-quinoline, *filaggrin*, drug development, atopic dermatitis

## Abstract

Atopic dermatitis (AD) is a common T-helper 2 (Th2) lymphocyte-mediated chronic inflammatory skin disease characterized by disturbed epidermal differentiation (e.g., *filaggrin* (*FLG*) expression) and diminished skin barrier function. Therapeutics targeting the aryl hydrocarbon receptor (AHR), such as coal tar and tapinarof, are effective in AD, yet new receptor ligands with improved potency or bioavailability are in demand to expand the AHR-targeting therapeutic arsenal. We found that carboxamide derivatives from laquinimod, tasquinimod, and roquinimex can activate AHR signaling at low nanomolar concentrations. Tasquinimod derivative (IMA-06504) and its prodrug (IMA-07101) provided full agonist activity and were most effective to induce *FLG* and other epidermal differentiation proteins, and counteracted IL-4 mediated repression of terminal differentiation. Partial agonist activity by other derivatives was less efficacious. The previously reported beneficial safety profile of these novel small molecules, and the herein reported therapeutic potential of specific carboxamide derivatives, provides a solid rationale for further preclinical assertation.

## 1. Introduction

During epidermal differentiation, multiple structural proteins are expressed in the last living cell layer, the *stratum granulosum*. Amongst others, filament aggregating protein (*filaggrin*), encoded by the *FLG* gene, is processed into the cornified envelope, an insoluble network consisting of the debris of keratinocytes—corneocytes—that are tightly crosslinked and imbedded in a matrix of lipid components ultimately forming the physical barrier of the skin: the *stratum corneum.* Disturbed epidermal differentiation and skin barrier function loss are key hallmarks of common chronic inflammatory skin diseases such as atopic dermatitis (AD) and psoriasis (Pso). The epidermal differentiation process is affected by the disease-specific cytokine milieu (the Th2-cytokines in AD (e.g., IL-4) or the Th1/Th17 cytokines in Pso), or due to genetic predisposition mostly associated to AD (e.g., *FLG* loss-of-function mutations [1,2], *FLG* copy number variation [3], *hornerin* (*HRNR*)- Single-Nucleotide Polymorphism (SNP) [4,5], *small proline-rich protein 3* (*SPRR3*) [6]. Most therapeutic strategies are aimed at general or targeted immunosuppression combined with indifferent skin moisturizing emollients. However, the direct targeting of the epidermal differentiation process in combination with immunomodulatory effects could be an attractive therapeutic avenue by killing two birds with one stone.

Evidence for such effective therapeutics came from our studies on the molecular mechanism of coal tar. Coal tar is a viscous liquid that is obtained by burning coal and is thought to consist of at least 10,000 chemicals of which many are characterized as polycyclic aromatic hydrocarbons (PAHs). Although topical application of coal tar is an ancient treatment option for AD and Pso, its exact therapeutic mechanism of action has long been unknown, until recently. PAHs in coal tar activate the aryl hydrocarbon receptor (AHR) in keratinocytes, thereby counteracting the keratinocyte activation towards AD-related interleukins and restoring the expression levels of affected differentiation proteins, including that of *FLG* [7]. In addition, genes encoding antimicrobial peptides are upregulated which together with restored differentiation capability and reconstituted skin barrier properties are thought to contribute to dampening of the inflammatory processes in skin [8]. 

The AHR is a highly conserved receptor and member of the family of basic helix–loop–helix transcription factors, that can be activated by a wide variety of both exogenous and endogenous ligands. Dioxins are a group of chemicals that are considered organic pollutants. TCDD (2,3,7,8-tetrachlorodibenzo-p-dioxin), one of the most widely used polychlorinated dioxins and considered an AHR model ligand, is often used to study AHR signaling in epidermal keratinocytes [9,10]. As TCDD is highly toxic with prolonged half-life of several years, dioxins such as TCDD are not suited for therapeutic purposes. Short-lived AHR ligands include UV-induced endogenous tryptophan metabolites [11] (hence the postulated role for AHR-mediated therapeutic effects of UV therapy in PSo). Other metabolites can be formed by members of the skin’s microbiome, and dietary plant constituents, as well as several pharmaceutics [12,13,14,15]. Upon activation, AHR signaling is known to regulate a plethora of cellular processes, e.g., embryonic development [16], keratinocyte proliferation and differentiation [17], epidermal barrier formation [14,18], immune cell development [19], angiogenesis [20], and many more [16,18,19,21,22,23,24]. However, studies also report contradicting outcomes depending on the tissue and disease context [25], and AHR ligand promiscuity has long been a subject of research [26]. 

Ever since the working mechanism of coal tar via AHR was elucidated, a search for novel AHR targeting therapeutics has been ongoing. This has led to the development of tapinarof, a natural AHR ligand that is currently in Phase 3 trials as a topical drug for AD and PSo [27,28]. However, other immunomodulatory molecules that are investigated in clinical trials for different indications, such as quinoline-3-carboxamide derivatives, also act on the AHR signaling pathway [29]. Laquinimod (LAQ) targets the AHR and is effective in alleviating disease symptoms in experimental models of multiple sclerosis and Huntington’s disease [30] and reduces IL-17 levels [31,32]. Roquinimex (ROQ, the first clinical compound in the series) and tasquinimod (TASQ) have been investigated being potential immunomodulating drugs effective in cancer treatment and in autoimmune diseases. Interestingly, ROQ was found effective against Pso in two patients that were included in a phase 2 clinical renal cell carcinoma trial [33]. Although LAQ, TASQ and ROQ may lead to AHR activation, they do not activate AHR signaling in their original form. Instead, intracellular metabolism generates potent N-dealkylated metabolites that are capable of AHR binding, as previously described in the patent application [34]. These N-dealkylated compounds have been tested in toxicity studies in vitro [35] and as diacetyl prodrugs in vivo [36] at high doses with minor signs of adverse effects. Albeit the structural similarity to TCDD and shared theoretical receptor binding modes [35], these carboxamide derivatives are considered less metabolically resistant as compared to TCDD. The pharmacokinetic clearance of AHR ligands would be important to mitigate the adverse effects of sustained AHR signaling.

In this study, carboxamide derivatives from LAQ, TASQ and ROQ were analyzed for their AHR activating potential in human keratinocytes and in human epidermal models. Primary read out was the expression level of epidermal differentiation genes and proteins. In addition, the ability to rescue deprived *FLG* expression, and other important epidermal barrier proteins in AD-like organotypic skin models, was investigated to identify new drug candidates for further preclinical testing.

## 2. Results

### 2.1. Structure of IMA-Compounds

LAQ and TASQ are both metabolized by CYP3A4 (N-dealkylation) to form the AHR-active metabolites IMA-06201 and IMA-06504 in low concentrations (Figure 1A,B, Appendix A). These AHR-active compounds have extremely low aqueous solubility and are therefore not appropriate for in vivo or clinical use. Even though prodrugs IMA-08401 and IMA-07101 also have low aqueous solubility, they can easily be formulated (e.g., PEG-400) [36]. Furthermore, the in vivo hydrolysis of the prodrug IMA-08401 results in higher levels of IMA-06201 than corresponding levels from LAQ metabolism (unpublished). Analogously to IMA-06201 and IMA-06504, compound IMA-05101 is an AHR agonist and a metabolite of the clinical compound ROQ. IMA-01403 was synthesized by blocking the IMA-05101 4-OH with a benzyl group, which adds extra bulk and disrupts internal hydrogen bonding, expected to severely reduce the activity of the compound (Figure 1C, Appendix A). 

### 2.2. IMA-Compounds Induce AHR Activity in Reporter Cell Line and Primary Keratinocytes

First, all newly synthesized derivatives were screened for AHR activating potential using the established human HepG2 (40/6) reporter cell line [37]. LAQ was used for comparison being a parent compound, 2,3,7,8-Tetrachlorodibenzo-p-Dioxin (TCDD) was included as a full AHR agonist at a saturating dose of 10 nM. All metabolites and prodrugs derived from LAQ, TASQ, and ROQ were able to activate AHR signaling in a clear dose dependent manner, maximally leading to 110–145% of the TCDD response. Blocking the -OH group in IMA-01403 indeed reduced the agonist activity (Figure 2A).

Next, we subjected primary human keratinocyte monolayer cultures to all derivatives. Again, clear dose dependent gene expression levels of AHR target genes, cytochrome P450 1A1 (*CYP1A1*) and 1B1 (*CYP1B1*) were observed (Figure 2B, Appendix A). Prodrug IMA-07101 and its active metabolite IMA-06504 (from TASQ) and prodrug IMA-08401 (from LAQ) exhibited saturated responses such as those seen after TCDD exposure. *AHRR* gene expression levels indicate the activation of a negative control feedback loop to downscale prolonged AHR activation, which was most evident for TCDD and the TASQ derivatives (Appendix A). Only *CYP1A1* mRNA expression levels reaching a 1000-fold change resulted in significantly induced enzymatic *CYP1A1* activity, as observed for TCDD, IMA-06504, and IMA-07101 (all at 10 nM) (Figure 2C). Cell viability was unaffected in all cell cultures as lactate dehydrogenase (LDH) levels were comparable to control (unstimulated) keratinocyte cultures (Figure 2D). 

### 2.3. AHR-Mediated Expression of Epidermal Differentiation

Given the stimulating effects of coal tar, TCDD, and other AHR ligands on epidermal differentiation through AHR-dependent mechanisms [7,10,38,39], we analyzed the following marker levels for terminal differentiation: *filaggrin* (*FLG*), *hornerin* (*HRNR*), *involucrin* (*IVL*), and *loricrin* (*LOR*) (Figure 3). Stimulation with IMA-07101 significantly and dose-dependently induced *FLG*, *HRNR,* and *IVL* levels, similar to TCDD. Based on these results, we subjected organotypic human epidermal equivalents (HEEs) to 1 nM IMA exposure during the final 96 h of the air–liquid interface culture. In particular, the increase in numbers of *stratum granulosum* layers was most apparent after IMA-06504, IMA-07101, and IMA-08401 exposure (Figure 4A) and correlated to the levels of *CYP1A1* expression (Figure 4B). We also observed epidermal thickening and induction of *stratum corneum* layers most notably after AHR activation via TCDD and the TASQ derivatives. *FLG* protein expression levels were most strongly induced by TASQ derivative IMA-06504 and its prodrug IMA-07101 (Figure 4C). This finding was further substantiated by *loricrin* (*LOR*) and *involucrin* (*IVL*) immunostainings of IMA-07101 exposed HEEs (Appendix A), and subsequent semi-quantitative analysis (Appendix A). Thus, activation of AHR signaling by LAQ and TASQ derivatives resulted in similar stimulating effects on epidermal development and formation as previously described for coal tar [7].

### 2.4. Therapeutic Effect of IMA-Compounds in Organotypic AD-like Epidermal Models

The potential of IMA-compounds to counteract detrimental effects of Th2 cytokines (e.g., IL-4) on epidermal differentiation protein expression, as seen for coal tar treatment, was investigated in an AD-like disease model. Hereto, HEEs generated from human primary keratinocytes harboring a heterozygous *FLG* mutation (Appendix A) were exposed to 10 ng/mL IL-4 for 96 h in total. After the first 24 h (disease initiation phase), the AD-HEEs were additionally stimulated with 1 nM of IMA-06504, IMA-07101, IMA-01403 (as a negative control) and TCDD for the final 72 h (treatment phase). IL-4 treated HEEs present with a thickened epidermis with fewer granular layers. Treatment with the TASQ derivatives increased the number of granular layers (Figure 5B). AHR activation in the AD-model by the TASQ derivatives and TCDD was verified by *CYP1A1* protein expression detection (Appendix A). The IL-4 mediated downregulation of *FLG*, *LOR*, and *IVL* expression, indicated by irregular staining patterns and staining of less epidermal layers, was effectively counteracted by IMA-07101 and its active metabolite IMA-06504. IMA-01403 was ineffective (Figure 5B). These findings were substantiated by semi-quantitative analysis (Appendix A). 

## 3. Discussion

In search of novel therapeutic AHR ligands for the treatment of inflammatory skin diseases, we here showed that carboxamide derivates are full AHR agonists in the low nanomolar range. TASQ metabolites most effectively induce *FLG* and other epidermal proteins important for skin barrier function, both in normal skin conditions and in a Th2-cytokine dominated inflammatory milieu representing AD-like inflammation. 

The first compound from the carboxamide compound class, ROQ (drug name Linomide) was withdrawn from clinical trials due to severe drug-related adverse events in phase III trials with MS patients. [40,41,42]. Thereafter, LAQ and TASQ were developed and intended as safer derivatives [43,44]. Phase II/III clinical trials indicate therapeutic efficacy of LAQ in patients with multiple sclerosis [45,46], as also confirmed in recent meta-analysis [47]. Herein, a significantly higher risk of LAQ treatment was associated with back pain, headache, and vomiting [47]. Tasquinimod is intended for the oral treatment of prostate cancer, but also comes with adverse events (e.g., skeletal pain, digestive disorders, insomnia) [48,49]. Systemic exposure of exogenous AHR ligands may modulate and disrupt physiological AHR signaling given its expression in many tissues [22,50]. Topical application and thus local and tissue specific targeting in dermatological indications may therefore provide a better setting for the use of AHR ligands as therapeutics. Lead optimization of LAQ and TASQ resulted in increased potency of AHR activation in keratinocytes as LAQ only elevated AHR target gene levels at micromolar concentrations, whereas the IMA-metabolites induced AHR signaling already at 1 nM. This important step resulted in the first positive preclinical in vitro studies for dermatological indications herein described and provides solid ground for further preclinical development and topical formulation. 

The in vivo safety aspects of the IMA-compounds have been studied in rodents, where no clinical signs of subacute toxicity were observed upon systemic exposure [36], other than generic AHR-mediated responses not relating to dioxin-induced toxicity. IMA-06201 and IMA-06504 were classified as non-mutagenic in in vitro analysis [35]. Furthermore, IMA-compounds are predicted to be faster metabolized to inactive compounds as compared to TCDD, which may be supportive of an improved safety profile [35,36]. Of note, TASQ derivatives were not rapidly metabolized and inactivated in keratinocyte monolayer cultures, as indicated by the induced *AHRR* expression and *CYP1A1* enzyme activity after 48 h at similar levels as TCDD. These findings advocate for additional studies with prolonged culturing of keratinocytes after single dosing to determine the duration of AHR activation (e.g., *CYP1A1* expression dynamics) which may indicate TASQ metabolism rates and AHR ligand half-life as compared to dioxin. Our keratinocyte studies at least did not reveal cytotoxic effects (LDH leakage) or detrimental effects on the epidermal viability and morphology after exposure to IMA-compounds up to 96 h. However, also for TCDD, no acute cellular toxicity was observed, confirming the need for data on prolonged exposure to IMA-compounds, also including topical exposure in formulations rather than supplementation of culture medium as herein used.

The induction of epidermal differentiation by TASQ derivatives (even in the presence of IL-4) appeared more efficacious than for TCDD. This enables future dose reduction strategies for at least IMA-07101. Ideally, induction of epidermal differentiation should be retained while *CYP1A1* enzymatic levels are minimized. *CYP1A1* is associated with phase 1 metabolism and the generation of mutagenic epoxides from certain environmental pollutants. Although in vivo, CYP450 enzymes appear more important for detoxication than their activation of carcinogens [51]. Moreover, sustained *CYP1A1* activity may metabolize endogenous AHR ligands hence resulting in deprived physiological AHR signaling and skin tissue that is prone to inflammatory processes [51,52,53]. Besides concerns on phase 1 metabolism by *CYP1A1* due to AHR activation, a specific side effect, folliculitis, is reported in patients treated with topical AHR-activating therapies, such as coal tar and tapinarof [54,55]. The likelihood of other ligand classes, such as the IMA-compounds here investigated, causing similar side effects should be subject of further research.

TASQ derivatives IMA-06504 and IMA-07101 were most potent and effective for AHR-mediated induction of epidermal differentiation and the rescue of epidermal AD hallmarks by IL-4. The accelerated formation and development of the epidermis and skin barrier function upon AHR activation has also been shown upon in vivo TCDD treatment [10], in vitro coal tar treatment [7], and in Chinese traditional medicine [23]. Next to the regulation of terminal differentiation, AHR signaling was also reported to mediate keratinocyte proliferation [17,56]. Considering that proliferation and differentiation processes in both AD and Pso are disturbed, and that therapeutic effects of AHR activation are not specific to AD but are also demonstrated in Pso patients [57], efficacy of IMA-compounds in psoriasiform inflammation may be expected. Further preclinical studies are recommended including topical formulations and efficacy studies using other experimental AD models or ex vivo skin biopsies.

In future years, the search for novel or existing AHR-targeting drugs with high efficacies and minimized side effects will expand the currently limited arsenal of AHR-targeting therapeutics. Lead optimization of existing drug compounds, as we showed here for quinoline-3-carboxamide derivatives, is a promising strategy for the development of novel therapeutic AHR ligands to feed pharmaceutical pipelines. 

## 4. Materials and Methods

### 4.1. Synthesis of IMA-Compounds

Synthetic preparations of LAQ and IMA-compounds, except for IMA-01403, are described in patent application WO2012/050500A1 [34]. IMA-01403 was prepared by benzylation of N-(2,4-dimethoxybenzyl)-N-phenyl-1,2-dihydro-4-hydroxy-1-methyl-2-oxo-quinoline-3-carboxamide [34] using BnBr (1,5 eq.) and K_2_CO_3_ (2 eq.) in DMF at 60 °C overnight, followed by concentration and conventional workup. The crude product was deprotected (cleavage of N-2,4-dimethoxybenzyl) using cerium ammonium nitrate (CAN, 3 eq., 0,1 M in 95% aq. MeCN) at room temperature for 20 min, followed by concentration conventional workup and purification by silica chromatography (CH_2_Cl_2_) to give IMA-01403 in 50% overall yield.

^1^H NMR (400 MHz, CDCl_3_) δ 3.75 (s, 3H), 5.45 (s, 2H), 7.14 (t, 1H), 7.25 (d, 1H), 7.32–7.47 (m, 8H), 7.65 (t, 1H), 7.77 (d, 2H), 8.09 (d, 1H), 10.71 (s, 1H).

### 4.2. HepG2 (40/6) Luciferase Reporter Assay

Human HepG2 (40/6) AHR reporter cells (hepatocellular carcinoma liver cells) were seeded in a 24-well format in Minimal Essential Medium Eagle (α-MEM, Sigma-Aldrich, St. Louis, MO, USA) supplemented with 1% penicillin/streptomycin and 8% fetal bovine serum (125.000 cells in 500 µL). After 24 h, cells were stimulated with LAQ (100, 10, or 1 µM), IMA-compounds (10, 1, 0.1, or 0.01 nM), and TCDD (10 nM) for 4 h at 37 °C with 5% CO_2_. Thereafter, the cells were lysed in 200 mL lysis buffer (20 mM Tris-HCl, pH 7.8, 1% Triton X-100, 150 mM NaCl and 2 mM dithiothreitol) and stored at −80 °C. To detect luciferase activity, 20 µL cell lysate was mixed with 40 µL luciferase activity assay reagent (Promega Luciferase Assay system) and the luminescence signal measured (Synergy HT microplate reader, BioTek, Winooski, VT, USA).

### 4.3. Primary Keratinocyte Isolation

Surplus human skin was obtained from plastic surgery. Human primary keratinocytes were isolated as previously described and stored in liquid nitrogen (according to the principles of the Declaration of Helsinki) [58].

### 4.4. Monolayer Primary Keratinocyte Culture

Human primary keratinocytes were cultured submerged in 24-well plates at 37 °C with 5% CO_2_ using keratinocyte growth medium with required growth factors and supplements (KGM, Lonza; without antibiotics) until confluency was reached [59]. Keratinocyte differentiation was initiated by depletion of growth factors and the cells were simultaneously stimulated with the IMA-compounds (10, 1, or 0.1 nM), TCDD (10, 1, or 0.1 nM), and LAQ (100, 10, or 1 µM). Keratinocytes were harvested after 48 h of stimulation (re-stimulated after 24 h) and processed for RNA isolation and subsequent qPCR analysis. 

### 4.5. Lactate Dehydrogenase (LDH) Assay

The supernatant of the above-mentioned monolayer culture experiment was collected after 24 h of compound stimulation. Keratinocytes treated with 1% Triton X-100 in KGM were used as a positive control for 100% cell death. A cytotoxicity detection kit (Roche) was used to measure LDH activity according to the manufacturer’s protocol and absorbance was read at 490 nm with a microplate reader (Bio-Rad).

### 4.6. CYP1A1 Enzyme Activity Assay

The P450-Glo^TM^ *CYP1A1* assay system (Promega, Madison, WI, USA) was used to measure *CYP1A1* enzyme activity according to the manufacturer’s protocol. Keratinocytes were treated as described previously (see monolayer primary keratinocyte culture). After 48 h of stimulation (refreshed after 24 h), cells were washed with PBS after which 200 µL Luc-CEE substrate solution in KGM without growth factors was added to the wells and incubated for 3 h at 37 °C with 5% CO_2_. The culture medium was collected and mixed with luciferin detection reagent for detection of luminescence (Synergy HT microplate reader, BioTek, Winooski, VT, USA).

### 4.7. Human Epidermal Equivalent (HEE) Culture (Normal Skin and Atopic Dermatitis Model)

HEEs were generated according to the protocols previously described [58]. Briefly, 24-well cell culture inserts (ThinCert, Greiner Bio-One or Nunc, Thermo Fisher Scientific, Waltham, MA, USA) were coated with rat tail collagen (100 µg/mL, BD Biosciences, Franklin Lakes, NJ, USA) at 4 °C for 1 h. Then, 100.000–150.000 primary human keratinocytes were seeded submerged in 100–150 µL CnT-prime medium (CELLnTEC) After 48 h, cultures were switched to 3D differentiation medium, consisting of 60% CnT-Prime 3D Barrier medium (CELLnTEC) and 40% High Glucose Dulbecco’s Modified Eagle’s Medium (DMEM, Sigma-Aldrich, St. Louis, MO, USA). Then, 24 h later, the HEEs were lifted to the air–liquid interface (ALI), after which the differentiation medium was refreshed every other day.

In the normal skin model, HEEs were treated with the IMA-compounds (1 nM) and TCDD (1 nM) at day 4 of the ALI for 96 h (re-stimulation after 48 h). At day 8, HEEs were harvested and processed for qPCR and immunohistochemical analysis. 

To generate an atopic dermatitis (AD)-like (AD-HEE) model [7,60,61,62], human primary *FLG*^+/−^ keratinocytes (Appendix A) were used in combination with the disease-associated cytokine interleukin-4 (IL-4). AD-HEEs were first stimulated with IL-4 (10 ng/mL, Peprotech (supplemented with 0.05% bovine serum albumin, BSA, Sigma-Aldrich, St. Louis, MO, USA)) at day 4 of the ALI. After 24 h of disease initiation, HEEs were co-stimulated with the TASQ derivatives (IMA-06504 and IMA-07101), IMA-01403 and TCDD at a concentration of 1 nM and harvested 72 h later (re-stimulation after 48 h) for immunohistochemical analysis.

### 4.8. Immunohistochemistry

HEEs were fixed in 4% formalin for 4 h and processed for routine histology. The 6 µm paraffin sections were stained with hematoxylin and eosin (Sigma-Aldrich, St. Louis, MO, USA) after deparaffinization. For immunohistochemical analysis, sections were stained using an indirect immunoperoxidase technique with avidin–biotin complex enhancement (Vectastain, Vector Laboratories) using antibodies listed in Appendix A.

### 4.9. RNA Isolation Real-Time Quantitative PCR (RT-qPCR)

RNA was isolated with the Tissue Total RNA Kit (Favorgen, Vienna, Austria) according to the manufacturer’s protocol. RNA was treated with DNAseI (Invitrogen, Waltham, MA, USA) and used for cDNA synthesis using SuperScript IV VILO Master Mix (Invitrogen, Waltham, MA, USA) according to the manufacturer’s protocol. RT-qPCR analysis was performed using SYBR Green (Bio-Rad, Hercules, CA, USA). Target gene expression was normalized to the expression of the house keeping gene human acidic ribosomal phosphoprotein P0 (RPLP0) and relative expression levels were calculated by the ΔΔCt method [63]. Primer sequences (Biolegio, Nijmegen, The Netherlands) are depicted in Appendix A.

### 4.10. Quantification of Differentiation Protein Expression in HEEs

Image acquisition of HEE immunostainings was performed by a ZEISS Axio Imager equipped with a ZEISS Axiocam 105 color Digital Camera and a 40× objective. The ZEISS Axiocam 105 color is a compact 5-megapixel camera (2560 × 1920 pixels) for high resolution images with a 1/2.5” sensor. Two images per slide were chosen as representative for the whole culture and stored in CZI format. The images were analyzed with the cell image analysis software CellProfiler (Broad Institute) [64]. Software pipelines for *filaggrin* (*FLG*), *loricrin* (*LOR*), and *involucrin* (*IVL*) analysis were created (available upon request) and GraphPad Prism 9.0 was used for the visualization of the data. The quantified data is shown as the protein area occupied (µm^2^) per millimeter length of the epidermis. Because this value does not normalize for the thickness of the epidermis (differences in epidermal thickness were observed after compound stimulation), we also calculated the area of expression as percentage of the epidermal surface for comparison. 

### 4.11. Statistics

Statistical analyses were performed using GraphPad Prism 9.0 on experiments of at least *n* = 3 replicates. To determine statistical significance between multiple groups for the in vitro monolayer cultures, paired one-way analysis of variance (ANOVA; mixed effects model for repeated measures data (one missing value)) was performed followed by Tukey’s multiple comparison post hoc test. For the data obtained by CellProfiler software, the calculated protein area (µm^2^) per millimeter length was used to analyze significant differences between multiple groups by performing the Friedman test (paired nonparametric test), indicated as (*p* = x.xx) in the legend. If significant, Dunn’s multiple comparison post hoc test was executed. Each culture experiment (monolayer or organotypic HEE) includes biological donor replicates.

## Figures and Tables

**Figure 1 ijms-23-01773-f001:**
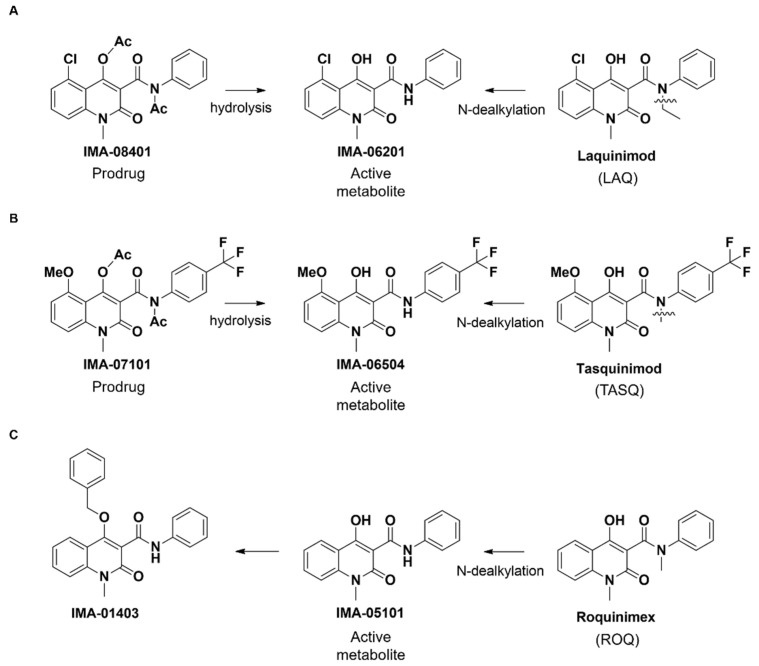
Structure of IMA-compounds. (**A**) Laquinimod (LAQ) is metabolized (N-dealkylation) to form the AHR-active metabolite IMA-06201. Hydrolysis of the prodrug IMA-08401 also forms the IMA-06201 derivative. (**B**) Tasquinimod (TASQ), its AHR-active metabolite IMA-06504, and the prodrug IMA-07101. (**C**) Roquinimex (ROQ), its AHR-active metabolite IMA-05101, and the 4-O-benzyl derivative IMA-01403.

**Figure 2 ijms-23-01773-f002:**
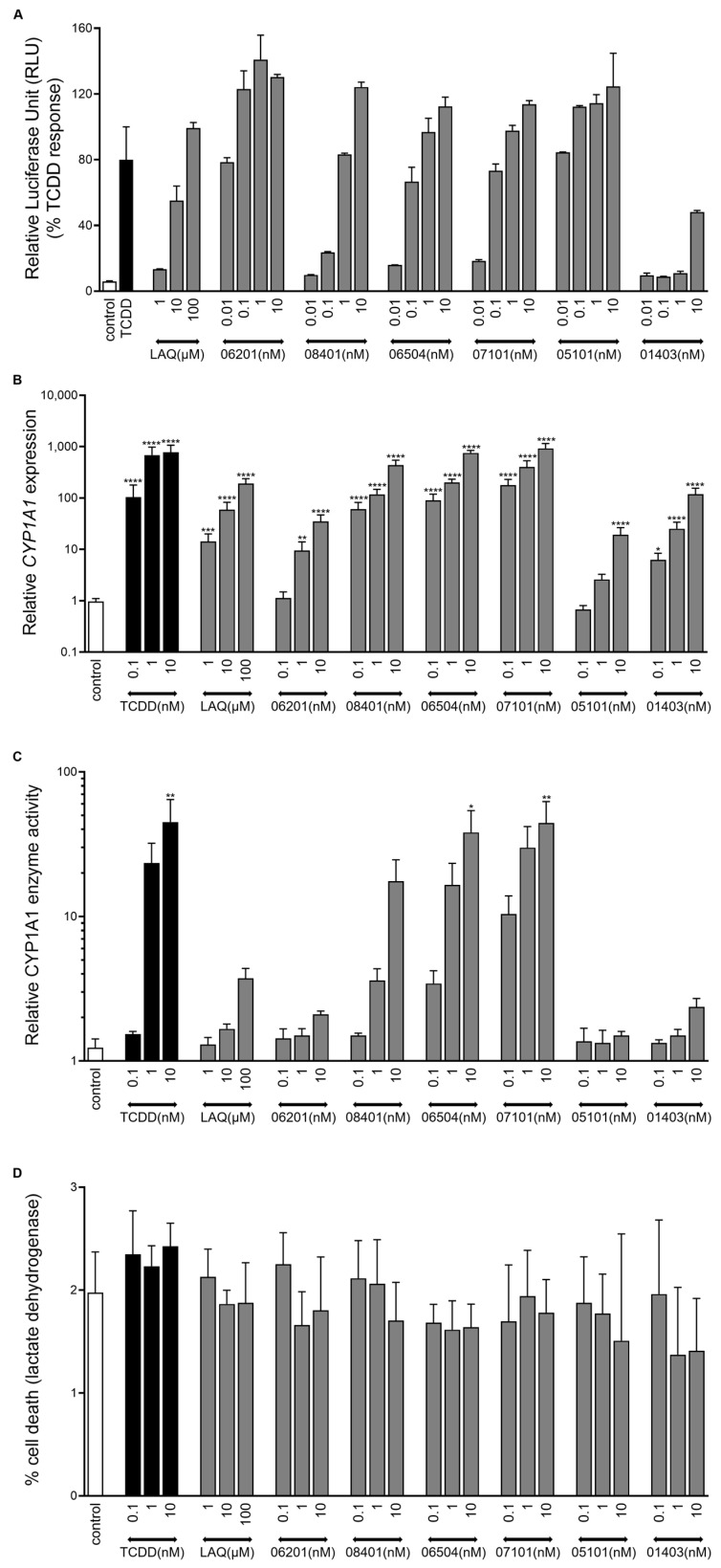
Induction of AHR activity by IMA-derivatives. (**A**) HepG2 40/6 cell cultures stimulated for 4 h with a concentration series of LAQ and IMA-compounds. TCDD (10 nM) was used as a full AHR agonist and set at 100% luminescent activity (*n* = 2). (**B**) *CYP1A1* mRNA expression levels, (**C**) *CYP1A1* enzyme activity (luminescent assay), and (**D**) percentage cell death (24 h measurement, lactate dehydrogenase (LDH) assay) of primary human keratinocytes (*n* = 3) stimulated for 48 h (re-stimulated after 24 h) with a concentration series of the IMA-compounds, LAQ, and TCDD. * *p* < 0.05, ** *p* < 0.01, *** *p* < 0.001, **** *p* < 0.0001. Mean +/− SEM.

**Figure 3 ijms-23-01773-f003:**
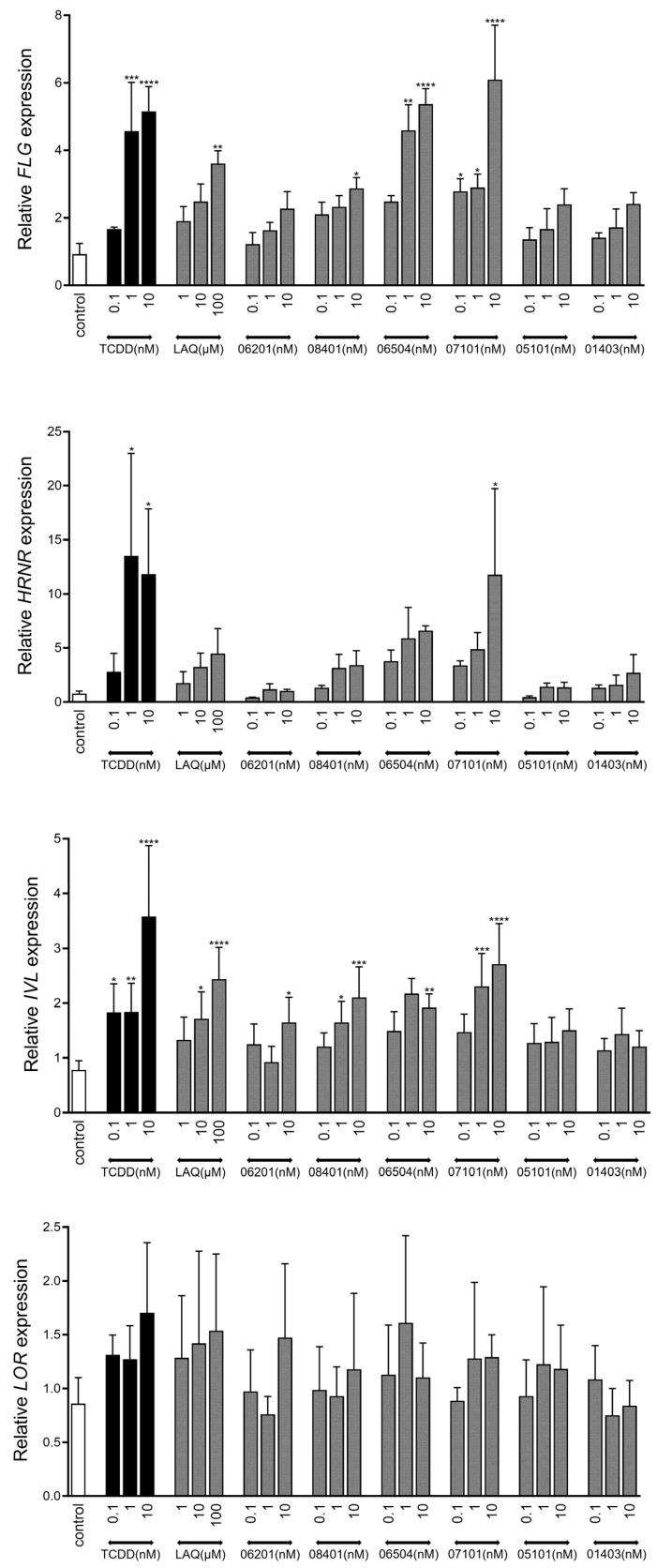
Upregulation of epidermal differentiation in vitro. Expression analysis of terminal differentiation genes *filaggrin* (*FLG*), *hornerin* (*HRNR*), *involucrin* (*IVL*), and *loricrin* (*LOR*) after 48 h stimulation (re-stimulation after 24 h) of monolayer primary human keratinocytes (*n* = 3) with a concentration series of the IMA-compounds, LAQ, and TCDD. * *p* < 0.05, ** *p* < 0.01, *** *p* < 0.001, **** *p* < 0.0001. Mean +/− SEM.

**Figure 4 ijms-23-01773-f004:**
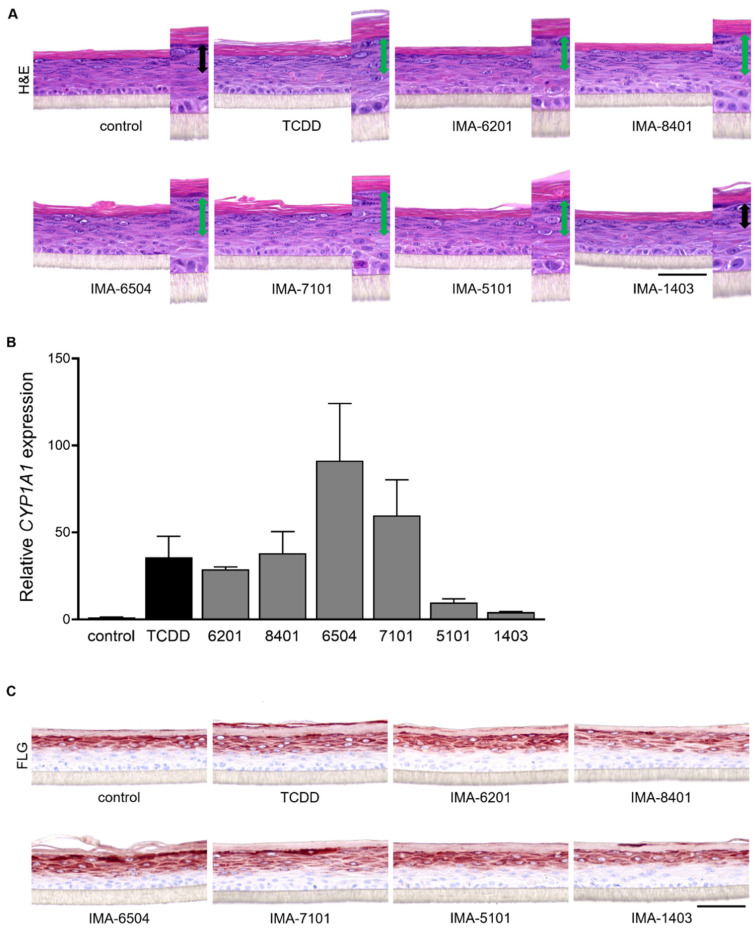
AHR activation and induction of epidermal differentiation in human epidermal equivalents. (**A**) H&E staining of human epidermal equivalents (HEEs) stimulated with 1 nM of IMA-compounds and TCDD for 96 h (re-stimulated after 48 h). Double headed arrows in magnified images indicate *stratum granulosum* layers (green arrows show an increase compared to the control-HEE). (**B**) *CYP1A1* mRNA levels (*n* = 2), mean +/− SEM. (**C**) Immunostaining for *filaggrin* (*FLG*) after 96 h of stimulation with IMA-compounds. Scale bar = 100 µm.

**Figure 5 ijms-23-01773-f005:**
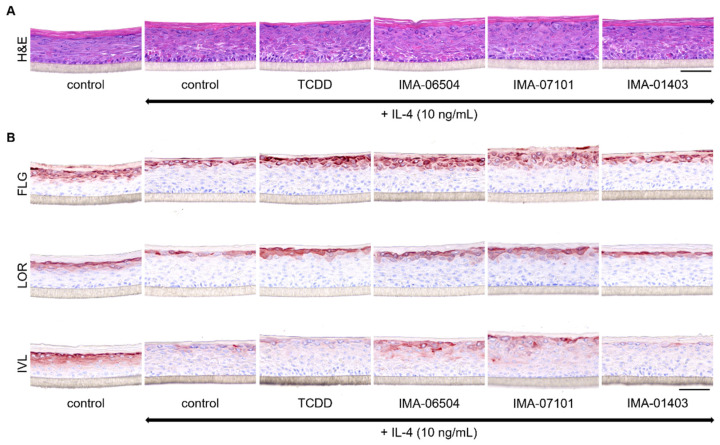
Therapeutic effect of TASQ derivatives in the AD-HEE model. (**A**) H&E staining of HEEs stimulated with 10 ng/mL IL-4 for 24 h followed by co-stimulation with the compounds for another 72 h. (**B**) Analysis of epidermal differentiation proteins via immunostainings for *filaggrin* (*FLG*), *loricrin* (*LOR*) and *involucrin* (*IVL*). Scale bar = 100 µm.

## Data Availability

The datasets used and analyzed during the current study are available from the corresponding author (E.H.v.d.B.) upon request.

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
