# Peer review of "Carboxamide Derivatives Are Potential Therapeutic AHR Ligands for Restoring IL-4 Mediated Repression of Epidermal Differentiation Proteins"

_ijms, 2022, doi:10.3390/ijms23031773_

Round 1
Reviewer 1 Report
Dear Professor Bogaard,
Congratulations on the present work. Your team produced a cool piece of science.
As you reported, the main goal for this work was to study the LAQ, TASQ and ROQ derivatives as potential activators of AHR in human keratinocytes and in human epidermal models.
In my opinion the manuscript is clear and well-organized. The report will be welcome.
Here are my suggestions:
1 - As the methods of IJMS are after the results section, I would like to suggest to mention the patent of Lars Pettersson in the introduction, as previously work.
2 - Why did You choose HepG2 as a reporter cell line? A skin' cell line would not be more appropriate?
3 - One of the core of your work is the isolation of LAQ, TASQ and ROQ derivatives.
The patent allows You to briefly describe it?
A better description of experimental control should increase the understanding of results. e.g. a mock was necessary?
4 - As we know Th2 response on inflamed skin is complex. In line 180 to 194, You checked the IMA-compounds to counteract detrimental effects of Th2 cytokines.
Why only the TASQ derivatives were used on AD-HEEs?
Moreover, I suggest You to include references showing that stimulation with IL-4 could represent Th2 response.
Finally, I also suggest a modification of the title by suppressing the word therapeutic and changing the Th2-cytokine to IL-4.
5 - line 322 mistyping.
Regards!
Reviewer 2 Report
The article of Gijs Rikken et al "Carboxamide derivatives are potential therapeutic AHR ligands 1 for restoring Th2-cytokine-mediated repression of epidermal 2 differentiation proteins" is a well-methodologically developed study of LAQ, TASQ, and ROQ metabolites as AHR-targeting therapeutic agents, considering their genetic involvement, enzymatic, immunologic and cellular effect levels. AHR is known to be involved in multiple physiological processes, and its response to the endogenous as well as exogenous ligands is highly controversial. In this regard, this kind of thorough study of the action mechanisms of AHR-targeted compounds, their derivatives, and metabolites are extremely relevant and are of interest both from a practical and theoretical point. The results are well documented, statistically properly processed, and accurately presented. However, there are small points that should be corrected:
- all abbreviations (as for example on lines 208, 41, 42, etc.) should be deciphered when first mentioned
- the term "antimicrobial genes" (line 54) is extremely unfortunate, it should be clarified what is meant.
Reviewer 3 Report
In this paper entitled “Carboxamide derivatives are potential therapeutic AHR ligands for restoring Th2-cytokine mediated repression of epidermal differentiation proteins”, the authors reported that carboxamide derivatives induce epidermal differentiation through aryl hydrocarbon receptor (AHR) – dependent mechanisms.
The study suggests therapeutic potential for atopic dermatitis (AD) in the near future.
Before a preclinical trial begins, I would suggest in vivo studies using AD model mice and drug safety data should be collected.
